# MHC matching improves engraftment of iPSC-derived neurons in non-human primates

Asuka Morizane[1], Tetsuhiro Kikuchi [1], Takuya Hayashi[2], Hiroshi Mizuma [2], Sayuki Takara[2], Hisashi Doi[2], Aya Mawatari[2], Matthew F. Glasser[3], Takashi Shiina[4], Hirohito Ishigaki[5], Yasushi Itoh[5], Keisuke Okita [6], Emi Yamasaki[1], Daisuke Doi[1], Hirotaka Onoe[2,7], Kazumasa Ogasawara[5], Shinya Yamanaka[6,8] & Jun Takahashi [1,9]

The banking of human leukocyte antigen (HLA)-homozygous-induced pluripotent stem cells (iPSCs) is considered a future clinical strategy for HLA-matched cell transplantation to reduce immunological graft rejection. Here we show the efficacy of major histocompatibility complex (MHC)-matched allogeneic neural cell grafting in the brain, which is considered a less immune-responsive tissue, using iPSCs derived from an MHC homozygous cynomolgus macaque. Positron emission tomography imaging reveals neuroinflammation associated with an immune response against MHC-mismatched grafted cells. Immunohistological analyses reveal that MHC-matching reduces the immune response by suppressing the accumulation of microglia (Iba-1+) and lymphocytes (CD45+) into the grafts. Consequently, MHC-matching increases the survival of grafted dopamine neurons (tyrosine hydroxylase: TH+). The effect of an immunosuppressant, Tacrolimus, is also confirmed in the same experimental setting. Our results demonstrate the rationale for MHC-matching in neural cell grafting to the brain and its feasibility in a clinical setting.

[1] Department of Clinical Application, Center for iPS Cell Research and Application, Kyoto University, Kyoto 606-8507, Japan. [2] RIKEN Center for Life Science Technologies (CLST), Hyogo 650-0047, Japan. [3] Department of Neuroscience, Washington University School of Medicine, St. Louis, M0 63110, USA. [4] Department of Molecular Life Science, Tokai University, School of Medicine, Kanagawa 259-1143, Japan. [5] Department of Pathology, Shiga University of Medical Science, Shiga 520-2192, Japan. [6] Department of Life Science Frontiers, Center for iPS Cell Research and Application, Kyoto University, Kyoto 606-8507, Japan. [7] Department of Neuroscience, Kyoto University Graduate School of Medicine, Kyoto 606-8507, Japan. [8] Gladstone Institute of Cardiovascular Disease, San Francisco, CA 94158, USA. [9] Department of Neurosurgery, Kyoto University Graduate School of Medicine, Kyoto 606-8507, Japan. Correspondence and requests for materials should be addressed to J.T. (email: jbtaka@cira.kyoto-u.ac.jp)

Cell therapy using pluripotent stem cells (PSCs) is considered a promising therapeutic application for many diseases[1] including Parkinson's disease (PD). After more than two decades of fetal neural cell grafting[2–4], cell therapy for PD with PSCs is expected to soon realize clinical application[5]. It has been shown that autologous transplantation is ideal from an immunological point of view[6–8]. Practically speaking, however, autologous transplantation at clinical grade that meets good manufacturing practice, quality assurance, and regulatory compliance is unlikely to become standard therapy due to its high cost and long preparation time per patient[9]. Another concern regarding autologous transplantation is the disease sensitivity of the donor cells from patients who have disease-specific genetic backgrounds. Allogeneic transplantation is therefore a preferred option. Major histocompatibility complex (MHC), or human leukocyte antigen (HLA) in humans, is expressed on the cell surface and is recognized by T-lymphocytes such that it plays a crucial role in the immune response after allogeneic transplantation. HLA haplotypes are determined by combinations of >16,000 HLA paternal and maternal alleles[10]. In organ transplantation such as kidney and bone marrow, matching HLA-A, -B, and -DR, improves the graft survival rates[11–13]. A recent report showed that MHC-matched allogeneic induced pluripotent stem cells (iPSC)-derived cardiomyocytes survived and functioned in monkeys that received immunosuppressive treatment[14]. These facts are consistent with the notion that HLA-matched transplantation using HLA-homozygous iPSCs could reduce immunological rejection[9, 15, 16]. For clinical application, such HLA-homozygous iPSCs would need to be banked. It is estimated that a tissue bank from 10 selected homozygote HLA-typed volunteers would match 37.7% of the population in the UK, and 150 similar volunteers would match 93%[15]. In other estimates, 50 lines would cover 90.7%[17] or 73%[16] of the Japanese population. For more than 30 years, allogeneic fetal neural cell grafting (HLA-mismatched transplantation) has been performed in clinical and basic studies for PD, and the neural grafts have shown good survival over a long time[3–5]. Yet some reports have suggested that immune responses to the neural grafts can affect graft survival and function[2, 3]. We therefore aimed to examine the effects of MHC matching in neural cell transplantation.

Here we show MHC matching reduces the immune response with microglia and lymphocytes, and increases the survival of iPSC-derived dopamine (DA) neurons in non-human primates (NHPs).

## Results

**Preparation of MHC-matched donor cells.** Our experimental design is described in Fig. 1a. Two iPSC lines, 1123C1-G and 1335A1, derived from cynomolgus macaques with homozygous MHC haplotypes (Mafa-HT1 and Mafa-HT4, respectively; Mafa is a cynomolgus macaque's MHC) were differentiated into DA neurons. The DA neurons were transplanted to monkeys in which at least one of the alleles was identical to the homozygotes for MHC-matched transplantation (16 animals in total. Figs. 1a and 2a; Supplementary Table 1, see also Methods section). This setting is referred to as the experimental model of HLA-matched allogeneic transplantation for PD. In the HT1 series of experiments (Cont#1–4 and Hetero#1–4), we used a donor cell line, 1123C1-G, to examine the effect of MHC matching. In the second HT4 series of experiments, we used another cell line, 1335A1, to confirm the results of the HT1 series and also to examine the effects of daily immunosuppression by Tacrolimus (Fig. 1). Donor iPSCs were maintained on iMatrix and constantly expressed pluripotent markers such as Oct4 and Nanog (Fig. 2b–f) with normal karyotype (Supplementary Fig. 1a, b). Both cell lines were differentiated into DA neural progenitors expressing several markers of the midbrain ventral mesencephalic phenotype, including Foxa2, Lmx1a, Nurr1, Corin, and MAP2, in a time dependent manner (Fig. 2g–m, Supplementary Fig. 1c–l). We transplanted the donor cells at differentiation day 28, and their major component was DA progenitor cell (TuJ1+/Foxa2+). The donor cells weakly expressed MHC class I, and interferon gamma (IFN-γ) stimulation increased its expression (Fig. 2n, o). On the other hand, the expression of MHC class II was below physiological level even with IFN-γ stimulation.

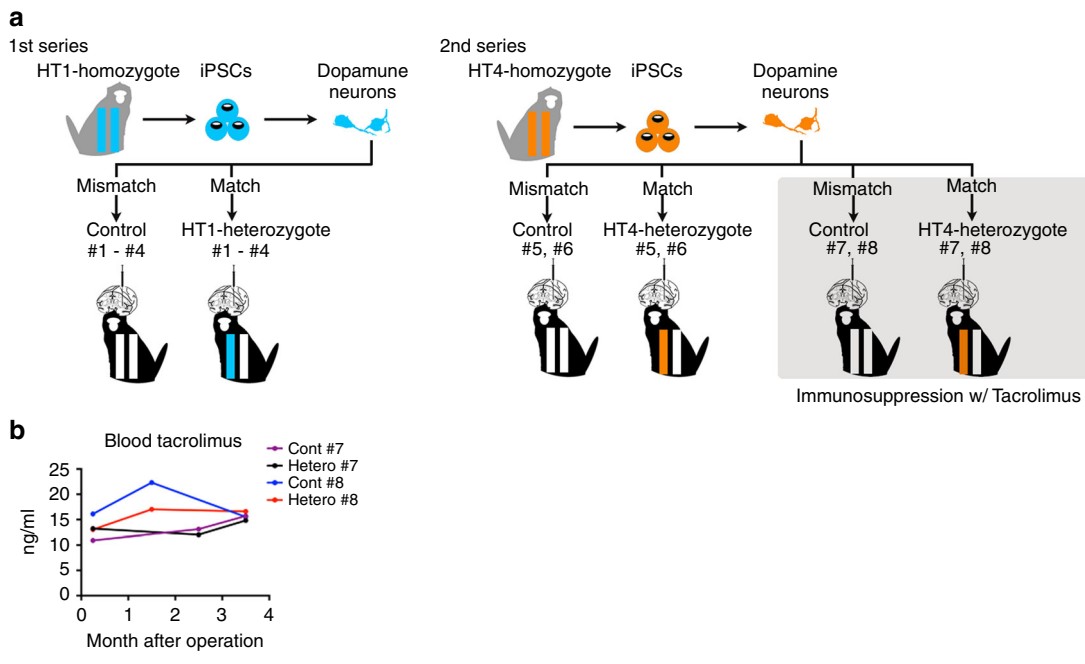

**Fig. 1** Experimental design. **a** Totally 16 animals received the grafts from two donor animal-derived iPSC lines. **b** Blood concentration of Tacrolimus was maintained at effective level

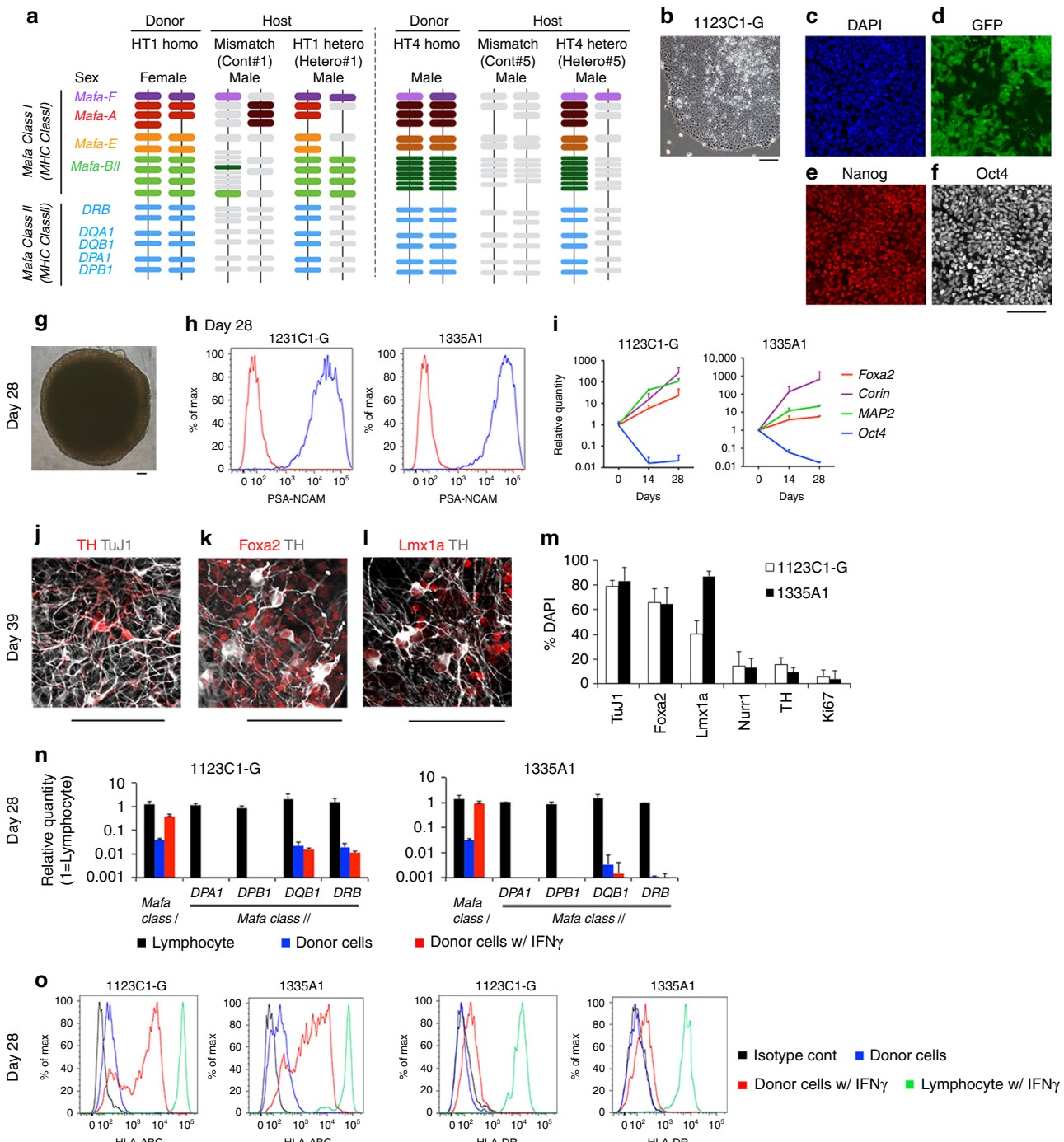

**Fig. 2** MHC genotypes of donors and recipients, and differentiation of MHC-homo iPSC-derived donor cells. **a** MHC genotypes of the donor and hosts in the first (HT1 haplotype) and the second (HT4 haplotype) experiments. Representative pairs of hosts are shown. See also Supplementary Table 1. **b–f** Characterization of MHC-homo iPSCs (1123C1-G). Phase contrast image (**b**), and fluorescent images (DAPI, **c**) with immunostaining for GFP (**d**), Nanog (**e**), and Oct4 (**f**). **g–m** Characterization of MHC-homo iPSC-derived dopamine neurons. An aggregate of donor cells at day 28 (**g**; phase contrast) contained more than 99% of cells positive for a neural marker, PSA-NCAM (**h**; flow cytometric analyses). **i** Quantitative PCR (qPCR) analysis of the cells during differentiation. **j–m** Immunostaining of iPSC-DA neurons at day 39. Quantification of immunocytochemical analysis for donor cells (**m**). **n**, **o** Expression of MHCs on the donor cells analyzed by qPCR (**n**) and flow cytometry (**o**). Scale bars: 100 μm (**b–g**, **j–l**), The data in **i**, **m**, **n** are shown as means ± SD (n = 4 independent experiments for **i** and 3 independent experiments for **m**, **n**)

**PET detected graft-induced neuroinflammation.** We grafted 5 million cells divided in six tracts to the left putamen (Fig. 3a). The grafts were detected as hypo- and hyper-intensity signals in T1-weighted and T2-weighted magnetic resonance imaging (MRI), respectively (Fig. 3b). We performed positron emission tomography (PET) imaging using two kinds of probes,

[11]C-PK11195 (Figs. 3c and 4) and (S)-[11]C-ketoprohen methyl ester ((S)-[11]C-KTP-Me; Fig. 5). These probes were short-lived, radio-labeled, and sensitive to specific biomarkers. Therefore, PET scanning allowed us to repeatedly and non-invasively monitor the concentration and location of the biomarker of interest. The two probes were expected to detect

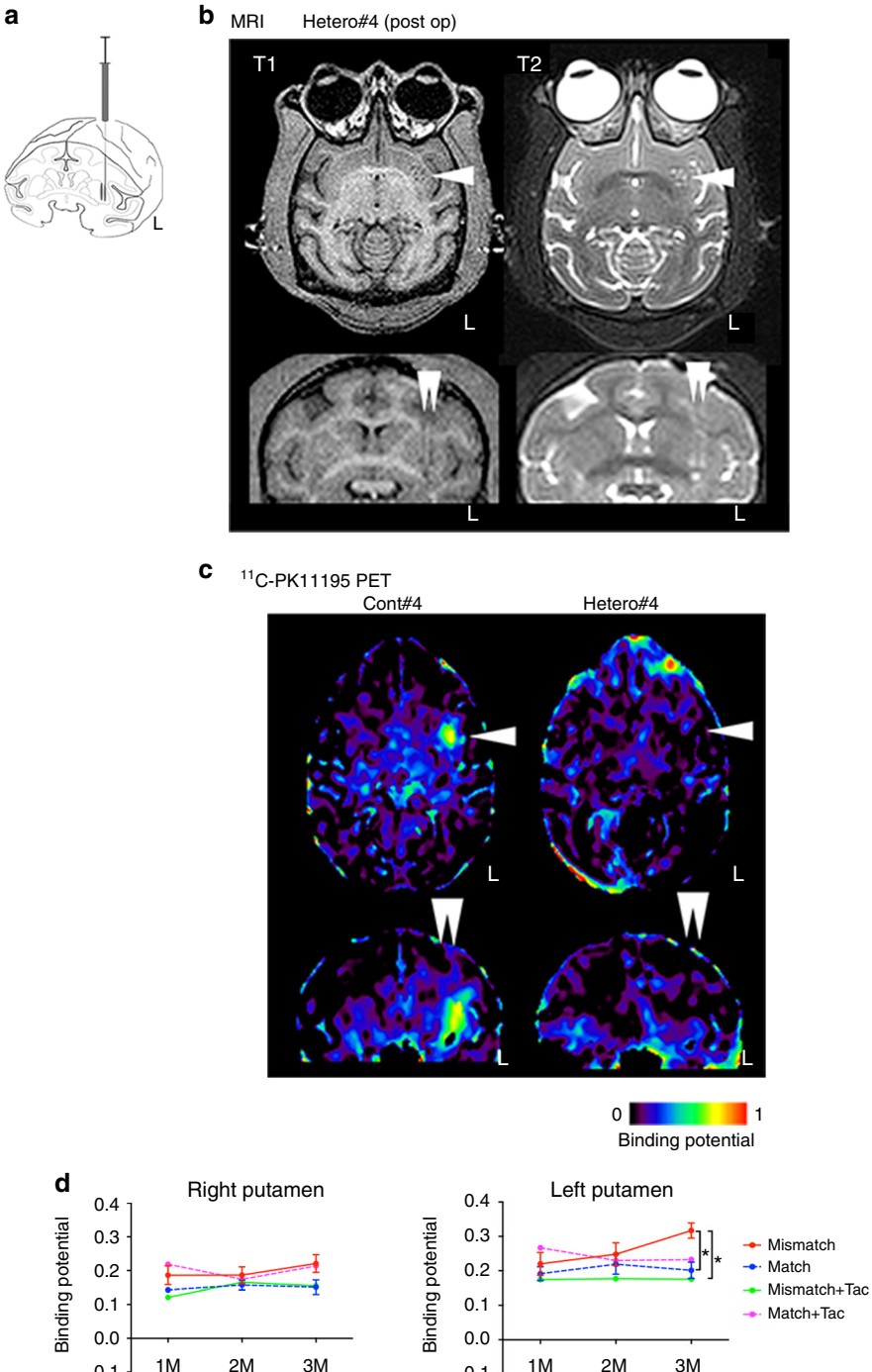

**Fig. 3** Live imaging of immune responses after cell transplantation. **a** Schema of grafts in the putamen of monkey brain. **b** Representative MRI after transplantation. Axial (*upper*) and coronal (*lower*) sections are shown with T1-weighted (T1, *left column*) and T2-weighted (T2, *right column*) images. **c** Binding potential (*BP*) images of $^{11}$C-PK11195 in positron emission tomography (PET) at 3 months after transplantation. Cont#4 showed increased BP in and around the grafted area. BP values ranging from 0 to 1 are presented with the *color bar*. **d** Chronological change in BP of $^{11}$C-PK11195 in the control (*right*) and grafted (*left*) sides of the putamen. The multiple comparisons test showed significance at 3 months in the *left* putamen between Mismatch vs. Match (*Bonferroni-corrected $P = 0.005$) and between Mismatch vs. Mismatch + Tac (*corrected $P = 0.006$). $n = 6$ for Mismatch and Match, $n = 2$ for Mismatch + Tac and Match + Tac, M: months after transplantation, Tac: Tacrolimus, *error bars* SEM, L: left side (**a–c**), *arrowheads* indicate grafted area (**b**, **c**)

neuroinflammation associated with an immune response by targeting translocator protein (TSPO)[18] and cyclooxygenase-1 (COX-1)[19], respectively. The chronological quantification of $^{11}$C-PK11195 binding potential (BP) in grafted putamen showed a potential inflammatory immune response for 3 months after transplantation (Figs. 3c, d and 4). Graft-host matching significantly affected the time course of BP in the grafted putamen

($F(1, 10) = 5.159$, $P = 0.046$, one-way repeated measure analysis of variance (ANOVA)), and MHC-mismatched grafts had BP that were ~50% higher ($0.317 \pm 0.022$; mean ± SEM) than those of matched grafts ($0.201 \pm 0.024$) at 3 months after transplantation (corrected $P = 0.005$) (Fig. 3d). We also confirmed the effect of graft-host matching was suppressed by Tacrolimus. There was a significant interaction between graft-host matching

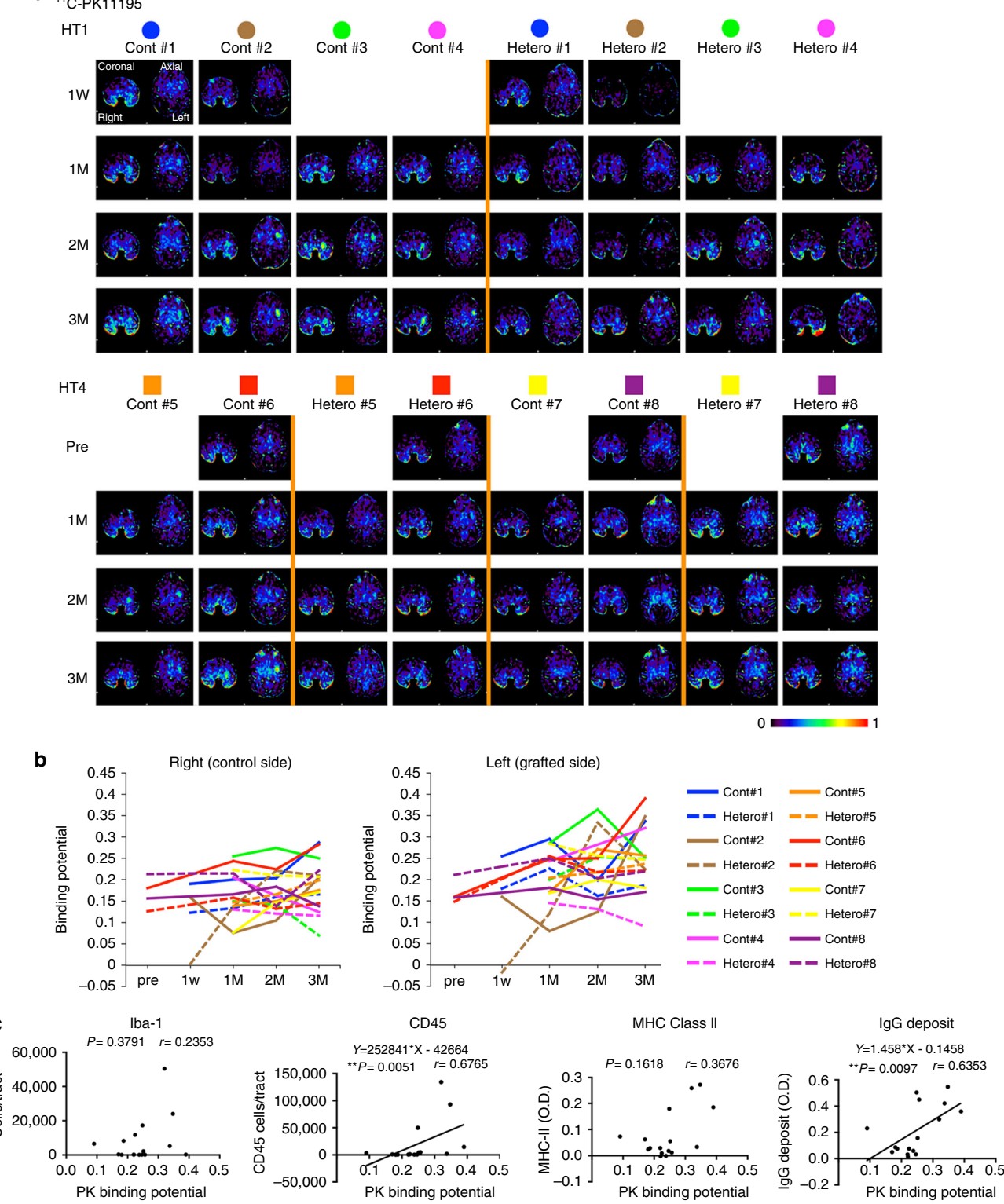

**Fig. 4** Binding potential (BP) images of $^{11}$C-PK11195 in positron emission tomography (PET). **a** Representative slices of BP images sectioned coronally and axially are shown for all animals and time points before (Pre) or after transplantation (1W, 1M, 2M, and 3M). **b** Chronological change of BP in the *left* (grafted side) and *right* (control side) putamen of each monkey. BP values ranging from 0 to 1 are presented with the *color bar*. **c** Spearman correlation analyses of $^{11}$C-PK11195 PET at 3 months and immunohistological results at 4 months. BP in each grafted putamen plotted vs. the average immunohistochemical results of Iba-1, CD45, MHC Class II, and IgG deposits from individual animals. Results from Spearman correlation analysis are given as *r*- and *P*-values in each graph. Tendencies of correlations are shown by linear regression lines for CD45 and IgG deposits. *M* months after transplantation, O.D.: optical density of the whole putamen, PK: $^{11}$C-PK11195, pre: preoperative analysis, W: weeks after transplantation

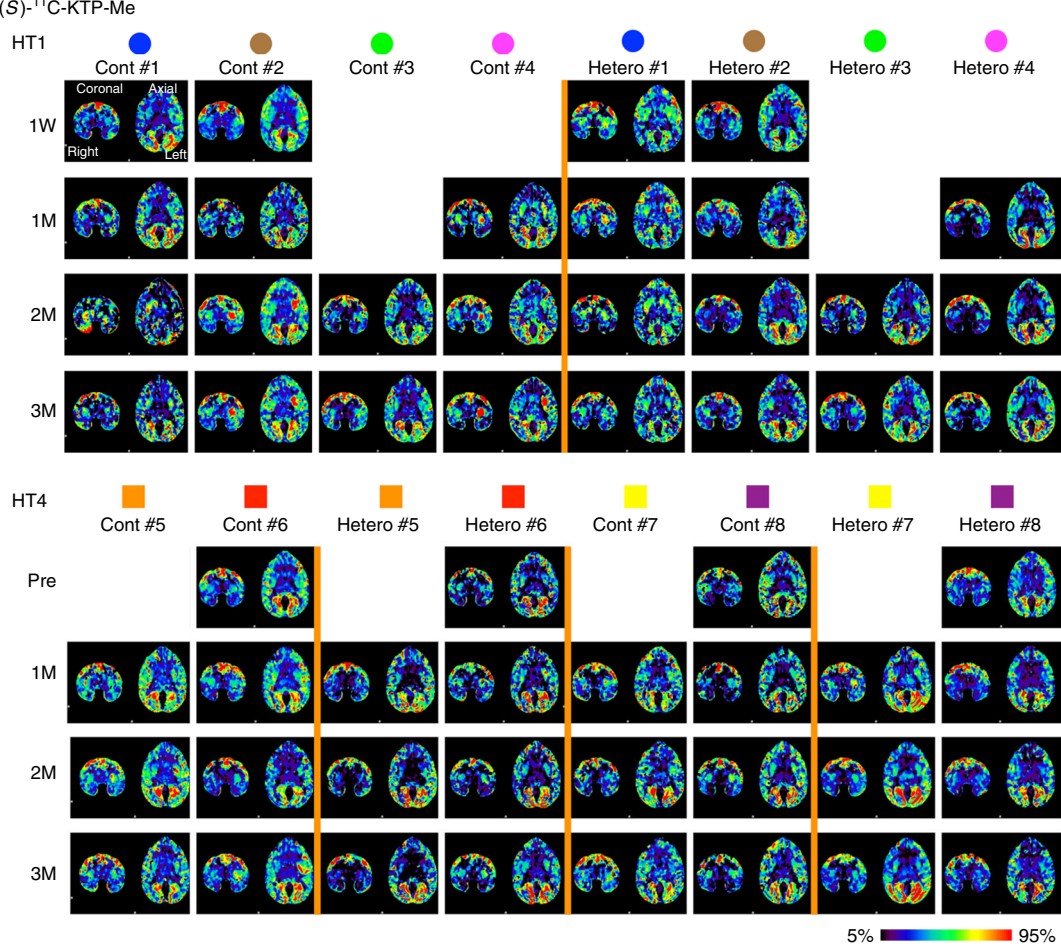

**Fig. 5** Binding potential (BP) images of (S)-$^{11}$C-KTP-Me in positron emission tomography (PET). BP images of (S)-$^{11}$C-KTP-Me PET in all animals and time points. For visualization, BP values are shown with a *color bar* ranging from 5th to 95th percentiles

and Tacrolimus ($F(1, 12) = 6.98$, $P = 0.022$, two-way repeated measure ANOVA), and when monkeys were treated with Tacrolimus, the BP of the mismatched graft at 3 months was decreased by ~50% ($0.176 \pm 0.036$) (corrected $P = 0.006$) such that it did not differ from the BP of matched grafts (Fig. 3d). A similar but insignificant pattern was observed in (S)-$^{11}$C-KTP-Me PET imaging, suggesting that the COX-1 is also involved in the inflammatory immune response to MHC-mismatched grafts (Fig. 5).

**Histological analyses at 4 months after transplantation.** For survival of the grafted cells in allogeneic transplantation, acute and sub-acute immune responses are critical and usually occur at 2–3 months after transplantation[10]. Therefore, we performed histological examination at 4 months after transplantation. An immunological response was indicated by the accumulation of host-derived activated microglia expressing Iba-1 and MHC class II and the infiltration of CD45+ leukocytes in the grafts (Figs. 6 and 7). More than 99% of the donor cells were committed to neural lineage before transplantation (Fig. 2h), and no GFP+ grafted cells expressed Iba-1 or CD45 in vivo (Fig. 6h, i), indicating that immune-responsive cells were derived from the host brain. Besides a cellular immune response, a humoral immune response was deduced from IgG deposits around the grafts. Statistical analysis showed significantly less Iba-1+ cell accumulation ($P < 0.0001$, $n = 24$ tracts in the HT1 series, paired $t$-test;

$P = 0.0005$, $n = 12$ tracts in the HT4 series, one-way ANOVA with Tukey's multiple comparisons test; $P = 0.004$, $n = 6$ animals in the HT1 and HT4 series, paired $t$-test, respectively) and less CD45+ leukocyte infiltration ($P = 0.029$ in the HT1 series, paired $t$-test; $P < 0.0001$ in the HT4 series one-way ANOVA with Tukey's multiple comparisons test; $P = 0.023$, $n = 6$ animals in the HT1 and HT4 series, paired $t$-test, respectively) in the MHC-matched grafts than MHC-mismatched ones (Fig. 7a–f). The BP of $^{11}$C-PK11195 in the PET study correlated with histological findings (Fig. 4c). The majority of infiltrating CD45+ cells was CD3+ T cells, of which 60–80% were CD4+ helper T cells and the rest CD8+ cytotoxic T cells (Fig. 6g). Immunosuppression with Tacrolimus in MHC-mismatched transplantation avoided the infiltration of CD45+ lymphocytes (Figs. 6b and 7d, h). Mesencephalic DA neurons were detected in the grafts that were positive for TH, Foxa2, and Girk2 (Fig. 8a–e). Statistical analysis showed more TH+ cells survived in the MHC-matched grafts compared with MHC-mismatched ones in both the HT1 and the HT4 series (Fig. 8f, g, $P = 0.004$, $n = 24$ tracts, paired $t$-test; $P = 0.0001$, $n = 12$ tracts, one-way ANOVA with Tukey's multiple comparisons test, respectively). The density of mature DA neurons (TH+ cells/mm$^3$) that survived was significantly higher in MHC-match grafts than MHC-mismatch ones (Fig. 8l, $P = 0.008$, $n = 6$ animals, paired $t$-test). In contrast, there was no significant difference in the density of surviving Foxa2+ cells (Fig. 8m, $P = 0.41$, $n = 6$ animals, paired $t$-test), suggesting that mature DA neurons (TH+, Foxa2+) are more sensitive to

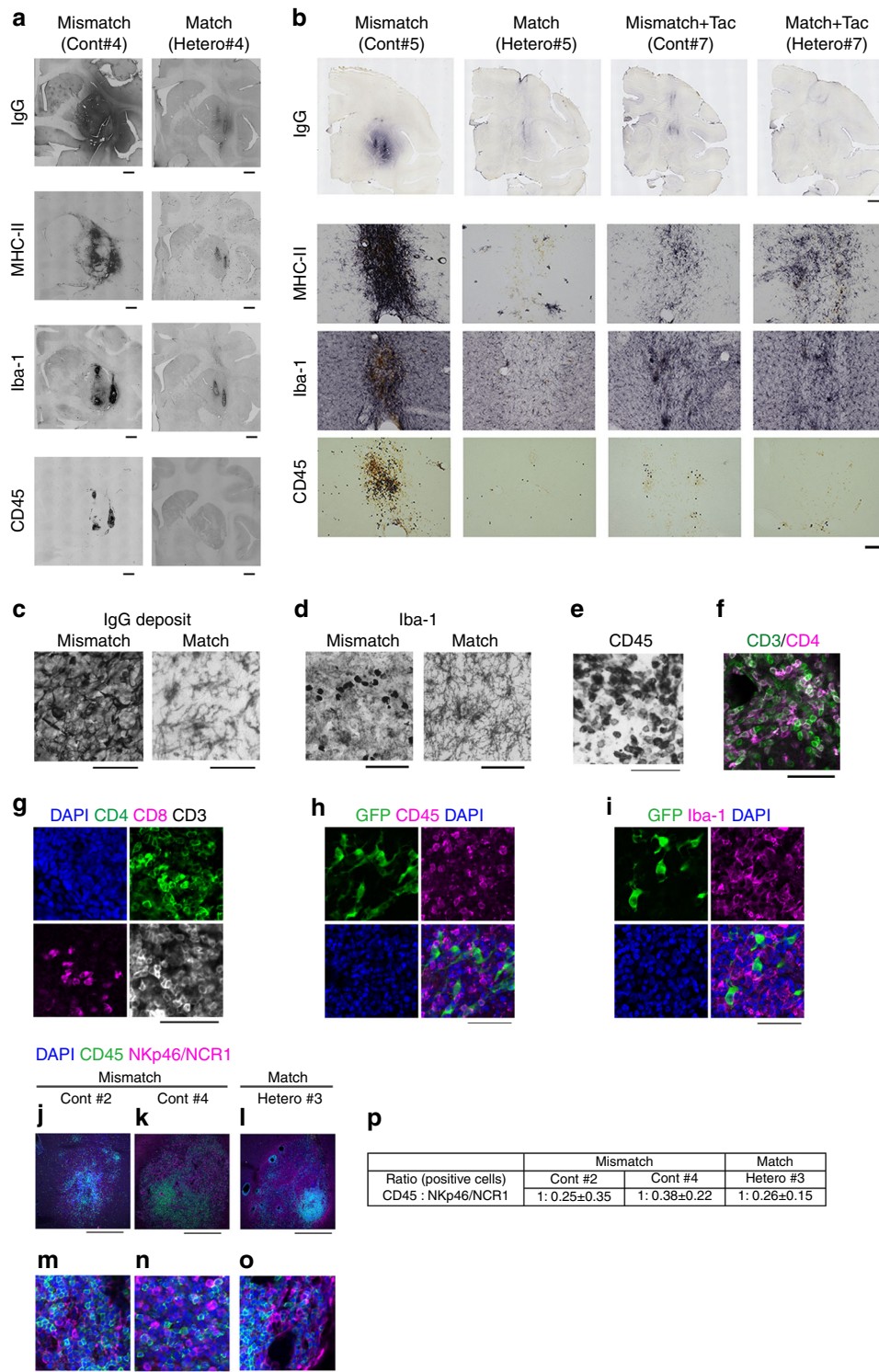

**Fig. 6** Histological analysis detected less immune response in MHC-matched transplantation. **a** Representative low magnified views of staining (Cont#4 and Hetero#4 from the HT1 experiments) at 4 months after transplantation. **b** Staining of Cont#5, Hetero#5, Cont#7, and Hetero#7 from the HT4 experiments. Representative view of grafts stained with DAB-Ni immunostaining for IgG, MHC-II, Iba-1, and CD45 are shown. **c–f** Magnified representative view of IgG deposits (**c**), Iba-1 (**d**), and CD45 (**e**) DAB-Ni staining in the grafts. Pictures are from Mismatch#4 (**c–e**) and Match#4 (**c**, **d**). **f** Representative view of immunostaining for CD3 (*green*) and CD4 (*magenta*) in the accumulation of lymphocytes in Mismatch#4. **g**, **h** Analyses of infiltrated lymphocytes. The majority of lymphocytes (CD3+) were positive for CD4 (helper T cell, *green*) and some were positive for CD8 (cytotoxic T cell, *magenta*; **g**). CD45 positive lymphocytes were derived from the host without GFP expression (**h**). **i** Immunostaining of the graft with GFP and Iba-1. **j–p** Activated natural killer (NK) cells stained by NKp46/NCR1 antibody (*magenta*). Low magnification (**j–l**), high magnification (**m–o**), and quantification (**p**). *Scale bars*: 2 mm (**a**), 5 mm (**b**; IgG), 100 μm (**b**; MHC-II, Iba-1, and CD45), and 50 μm (**c–i**, **m–o**), 500 μm (**j–l**). Quantitative data are presented as means ± SD (**p**)

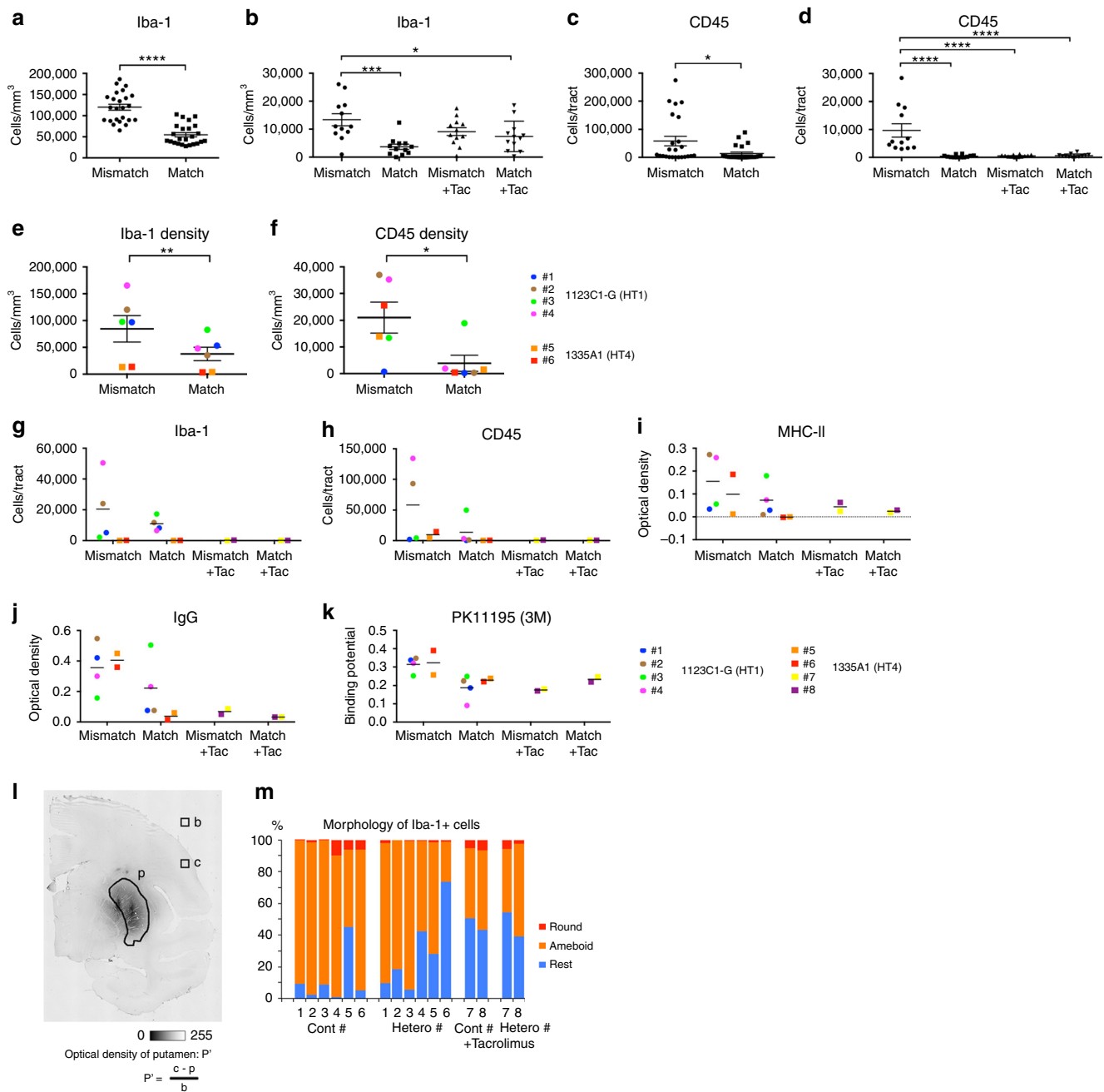

**Fig. 7** Quantitative analysis showed the effect of MHC matching on decreasing immune response after transplantation. **a–d** Quantitative analyses of Iba-1+ microglia (**a**, **b**) and CD45+ leukocytes (**c**, **d**) in the grafts; HT1 (**a**, **c**) and HT4 (**b**, **d**) experiments. **e**, **f** Averaged value plots of Iba-1+ (**e**) and CD45+ (**f**) cell densities in the grafts of individual monkeys. Combined data of Mismatch and Match from the HT1 and the HT4 series. **g–j** Averaged value plots in individual monkeys for the expressions of Iba-1 (**g**), CD45 (**h**), MHC class II (**i**), IgG deposits (**j**), [11]C-PK11195 (**k**). Mismatch (n = 6), Match (n = 6), Mismatch + Tacrolimus (n = 2), and Match + Tacrolimus (n = 2) experiments were analyzed. HT1 (#1–#4) and HT4 series (#5–#8) are separately shown in each graph. **l** The calculation method of optical density for MHC class II and IgG deposit staining. **m** Morphological analysis of Iba-1+ microglia. Resting microglia change their shape from ameboid to round upon activation. Data are presented as means ± SEM (n = 24 tracts for **a**, **c**; n = 12 tracts for **b**, **d**; n = 6 animals for **e**, **f**). Paired t-tests were performed for the HT1 experiment: ****P < 0.0001 (**a**), *P = 0.029 (**c**), **P = 0.004 (**e**), *P = 0.023 (**f**). One-way ANOVA with Tukey's multiple comparisons test was performed for the HT4 experiment: ***P = 0.0005 (**b**), *P = 0.0496 (**b**), ***P < 0.0001 (**d**). Tac: Tacrolimus, See also Fig. 6

immune responses than other types of grafted cells (TH−, Foxa2+). These results are consistent with previous reports that found DA neurons in substantia nigra pars compacta (SNc) are specifically sensitive to a variety of stresses such as mitochondrial oxidant stress[20] and mechanical stress[21]. We performed a classical one-way mixed lymphocyte reaction using peripheral blood mononuclear cells (PBMCs) to predict the immune

response, but found no significant correlation with the histological findings (Supplementary Fig. 2).

## Discussion

Although the brain is considered a less immune-responsive tissue[22], the present study clearly shows MHC matching can

reduce the number of Iba-1+ microglia and CD45+ lymphocytes and increase the number of surviving TH+ neurons in the grafts (Figs. 7a–d and 8f, g). One animal (MHC-match Hetero#3) showed a slight immune response (Fig. 7g–j). Possible

mechanisms for immune responses in this MHC-matched animal include an indirect pathway caused by minor antigens instead of MHC[23] or innate immunity caused by natural killer (NK) cells[24] and a complementary system. Although previous mouse

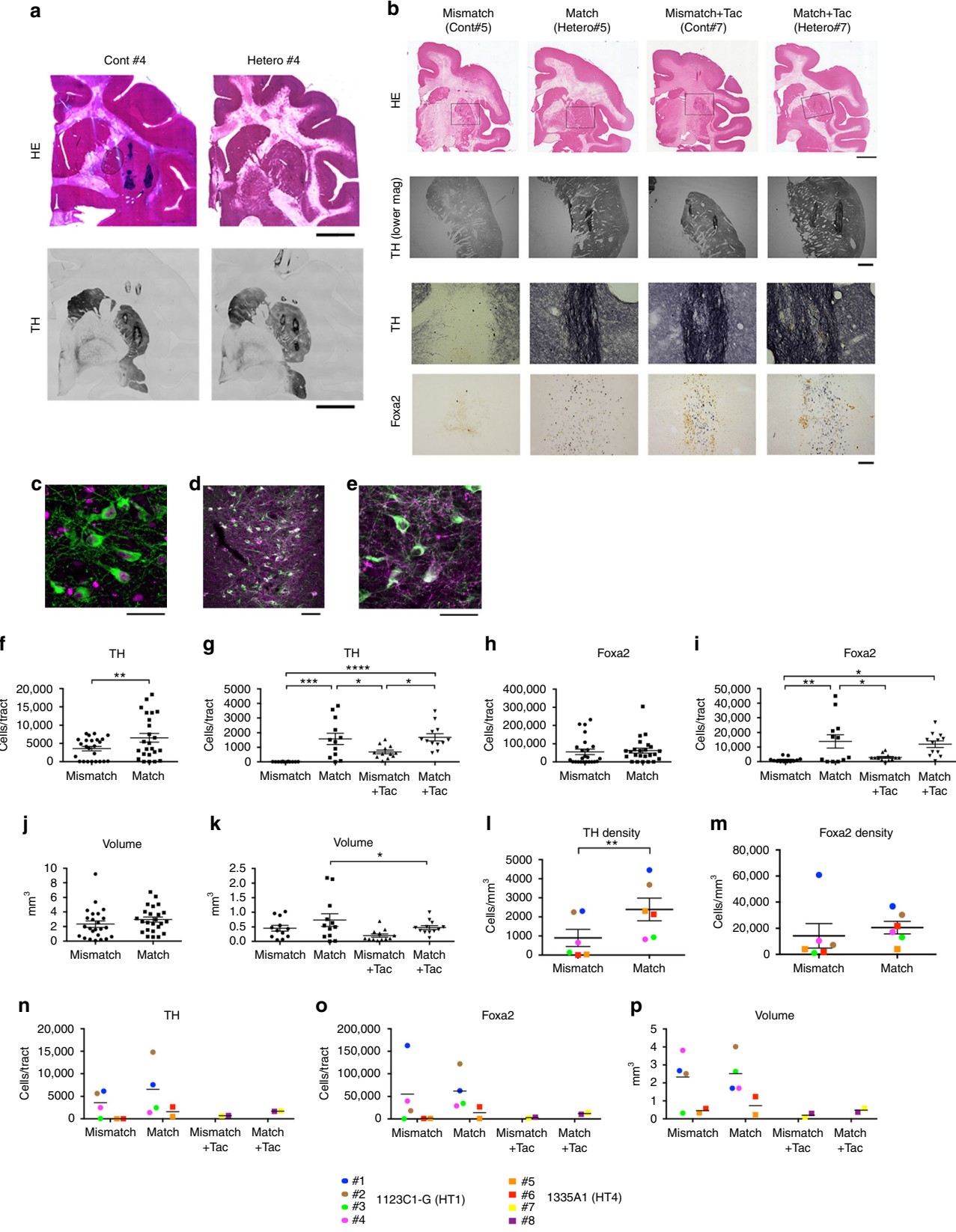

experiments reported that NK cells attack MHC-matched grafts more strongly than MHC-mismatched ones through a mechanism called hybrid resistance[25, 26], it is controversial whether the same mechanism exists in humans[27]. Furthermore, we found no difference in NK cell infiltration between the grafts of MHC-match Hetero#3 and MHC-mismatched Cont#2 and #4 (Fig. 6j–p). Additionally, Tacrolimus alone was effective at preventing the infiltration of lymphocytes into the grafts. It has been reported that MHC-mismatched grafts or human-derived xenografts survived and functioned in neurotoxin-lesioned PD monkeys with immunosuppression[28, 29]. In this study, we focused on acute and sub-acute immune responses so that we could perform histological analyses of immune-responsive cells. Even if controlled with MHC matching or immune suppressant, however, there is a chance that a late-onset immune response may occur. Therefore, further studies are needed to address the additional effects of MHC matching on synaptic formation and consequent functional recovery.

Although MHC matching was statistically effective at reducing the immune response, it did not completely evade the immune response even in the brain. Based on this finding, we propose that the combination of MHC matching and immunosuppression will provide the best strategy to control the immune response. Importantly, MHC matching may help to reduce the dose and duration of immunosuppressive drugs. Our study also indicates that PET studies would be helpful for detecting graft–host interactions and adjusting immunosuppression in clinical cases.

## Methods

**Non-human primates.** Sixteen purpose-bred male cynomolgus macaques (*Macaca fascicularis*) were used as the recipients, and two cynomolgus macaques were used as the donors. All animals were purchased from the Philippines archipelago (INA Research Inc.). There exists an isolated Filipino macaque population that has a limited variety of MHC polymorphisms compared with other populations. Hence, MHC homozygous and MHC heterozygous monkeys are frequently identified in this population[30]. The Mafa class I and II haplotypes of all animals were previously characterized by the Sanger sequencing method and high-resolution pyrosequencing[14, 30]. The donors were MHC homozygous. Four heterozygous male animals with HT1 haplotype and another four heterozygous animals with HT4 haplotype were used as recipients for the MHC-matched transplantation (MHC-Match; hetero). The other eight recipient animals did not have the same haplotype as the donors (MHC-mismatch; control) (Figs. 1a and 2a; Supplementary Table 1). In Hetero#7 and Hetero#8 monkeys, the alleles of the MHC class I, but not of MHC class II, were identical to the HT4 haplotype (Supplementary Table 1). Because the expression of MHC class II in the donor cells were below physiological level, those two monkeys were regarded as Hetero and were assigned to the group with immunosuppression. Because the monkeys used in this study were supplied from an isolated small colony and not from a closed colony like experimental rodents, the diversity of minor antigens might be less than of humans. On the other hand, it is known that the genetic polymorphism of cynomolgus monkeys has more variability than that of humans because of the longer evolutionary period (1.2 million years in cynomolgus monkey and 0.2–0.4 million years in human)[31, 32]. Furthermore, because this study focused on the immune response and survival of grafted cells, we adopted unlesioned host monkeys instead of neurotoxin-lesioned PD model, as it is difficult to prepare a set of equally lesioned PD monkeys due to the limited availability of MHC homozygous animals. All procedures for animal care and experiments were approved by the Institutional Animal Care and Use Committee of Animal Research Facility, CiRA, Kyoto University (Permission Number: 10-8-9), and performed in accordance with the Guidelines for Animal Experiments of Kyoto University, the Institutional Animal Care and Use Committee of Kobe Institute in RIKEN, and the Guide for the Care and Use of Laboratory Animals of the Institute of Laboratory Animal Resources (ILAR; Washington, DC, USA).

**Genotyping of MHC.** Total RNA was isolated from peripheral blood cells using TRIzol reagent (Invitrogen). complimentary DNA was synthesized using ReverTra Ace (TOYOBO) after treatment of the isolated RNA with DNase I (Invitrogen). A single MHC class I-specific primer pair in exons 2–4 (PCR product size: 514 or 517 bp) that could amplify all known MHC class I alleles and MHC class II locus-specific primer sets that included polymorphic exon 2 in Mafa-DRB (420 bp), Mafa-DQA1 (435 bp), Mafa-DQB1 (396 bp), Mafa-DPA1 (407 bp), and Mafa-DPB1 (333, 336, or 339 bp) were used for massively parallel pyrosequencing[14, 30]. After reverse transcriptase PCR amplification using high-fidelity KOD FX polymerase (TOYOBO), pyrosequencing of the PCR products was carried out using the GS Junior system and amplicon sequencing protocol (Roche). MHC genotypes were assigned by comparing the sequences to known MHC allele sequences at the Immuno Polymorphism Database (http://www.ebi.ac.uk/ipd/index.html).

**iPSCs.** To generate iPSCs, PBMCs from the donor female macaque were transfected with a combination of plasmid vectors (*OCT4, SOX2, KLF4, L-MYC, LIN28*, shRNA for *TP53*, and *EBNA1*)[6, 14]. The established primate iPSC lines 1123C1 and 1335A1 were maintained on iMatrix (Nippi, Inc., Japan) with AK01 or AK02 medium (Ajinomoto, Japan)[33]. After 10 passages (P10), 1123C1 was transfected with GFP to mark donor cells 1123C1-G. PB-EF1α-EGFP-IRES-PUROMYCIN plasmid vector was transfected into the cells using the FuGENE HD system, and puromycin was added to the medium to select the transfected cells (Fig. 2d). All iPSCs were used before passage 40. We used 1123C1-G for the first experiment and 1335A1 for the second experiment. For karyotype analysis, the iPSCs were treated with 0.2 μg/ml colcemid for 2 h. The cells were collected by Accumax, incubated in 0.075 M KCl for 20 min, and fixed with methanol and acetic acid (3:1). The chromosome spreads were prepared and examined as standard protocol with quinacrine mustard and Hoechst 33258 staining[14].

**Dopamine neuron induction.** For Cont#1, #2, #5–#8 and Hetero#1, #2, #5–#8, the primate iPSCs were differentiated into DA neurons using the modified SFEBq method[34] with dual SMAD inhibitors (Fig. 2h–k; Supplementary Fig. 1c–g)[35, 36]. For Cont#3, #4, and Hetero#3, #4, the iPSCs were differentiated into DA neurons using a modified protocol (Supplementary Fig. 1h–l) that was similar to the protocol for differentiating human iPSCs using an attachment culture on iMatrix[37]. We transplanted the cells at day 28 of the differentiation. For in vitro analysis, we dissociated the cells with Accumax (Innovative Cell Technologies) and replated them on eight-well glass chamber slides coated with iMatrix or sliced the spheres for immunostaining.

**Transplantation.** Floating aggregates (day 28) were harvested and dissociated into small clumps of 20–30 cells with Accumax. The cells were suspended in the last culture medium, which was Neurobasal medium with B27 and ascorbic acid (AA), dbcAMP, glial cell line-derived neurotrophic factor (GDNF), and brain-derived neurotrophic factor (BDNF). We also added a ROCK inhibitor, 10 μM Y27632, to increase the survival of the donor cells. The suspension was prepared at the concentration $2 \times 10^5$ cells/μl. Through a 22-gauge needle, we injected 4 μl of the suspension using a Hamilton syringe. We made six (two coronal × three sagittal) tracts of the injection in one side of the putamen. In total, $4.8 \times 10^6$ cells per animal ($8.0 \times 10^5$ cells/tract × 6 tracts) were injected into one side of the putamen according to the coordinates decided by the MRI of each monkey. The same volume of culture medium was injected to the contralateral side as the control. After surgery, antibiotics were given for 3 days. Monkeys Cont#7–8, Hetero#7–8 received a daily intramuscular immunosuppressant (Tacrolimus, 0.05 mg/kg; Astellas) from 1 day before the surgery until the day of killing to keep the blood

**Fig. 8** Histological analysis detected more dopamine neural survival in MHC-matched transplantation. **a, b** Representative low magnified views of staining at 4 months after transplantation. Cont#4 and Hetero#4 from the HT1 experiments (**a**). Cont#5, Hetero#5, Cont#7, and Hetero#7 from the HT4 experiments (**b**). Representative view of grafts with Hematoxylin and Eosin (HE) staining (**a, b**) and DAB-Ni immunostaining for tyrosine hydroxylase (TH; **a, b**) and Foxa2 (**b**). **c–e** TH (*green*), Foxa2 (**c**, *magenta*), and Girk2 (**d, e**, *magenta*) staining in the representative graft (Hetero#1). Magnified view (**c, e**) and lower magnified view (**d**). **f, g** Quantitative analyses of TH+ cells (**f, g**), Foxa2 (**h, i**), and graft volume (**j, k**) in the grafts; HT1 (**f, h, j**; n = 24 tracts) and HT4 (**g, i, k**; n = 12 tracts) experiments. **l, m** Averaged value plots of TH+ (**g**) and Foxa2+ (**h**) cell densities in the grafts of individual monkeys. Combined data of Mismatch and Match from the HT1 and the HT4 series. **n–p** Averaged value plots in individual monkeys for the expressions of tyrosine hydroxylase (TH, **n**), Foxa2 (**o**), and graft volume (**p**). Mismatch (n = 6), Match (n = 6), Mismatch + Tacrolimus (n = 2), and Match + Tacrolimus (n = 2). Scale bar: 5 mm (**a** HE and TH, **b** HE), 1 mm (**b**, TH lower mag), 100 μm (**b** TH and Foxa2, **d**), and 50 μm (**c, e**). Quantitative data are presented as means ± SEM (n = 24 tracts for **f, h, j**; n = 12 for **g, i, k**; and n = 6 for **l, m**). Paired t-tests were performed for **f, h, j, l, m**: **P = 0.004 (**f**) **P = 0.008 (**l**). One-way ANOVA with Tukey's multiple comparisons test was performed for **g, i, k**: ****P < 0.0001 (**g**), ***P = 0.0001 (**g**), *P = 0.049 (**g**, *left*), *P = 0.022 (**g**, *right*), *P = 0.025 (**i**, *upper*), **P = 0.006 (**i**), *P = 0.022 (**i**, *lower*), *P = 0.024 (**k**). Tac: Tacrolimus

concentration of the drug at 10–20 ng/ml (Fig. 1b). Under deep anesthesia, the animals were killed and perfused transcardially with 4% PFA after 4 months of observation.

**Immunohistochemistry**. For in vivo studies, fixed frozen brains were sliced at 40 μm thickness. The slices were immunologically stained using the free-floating method. The primary antibodies used are listed in Supplementary Table 2. For staining with CD3 and CD4 antibodies, we took an additional step for antigen retrieval using 10 mM citrate buffer at 100 °C for 40 min. Fluorescent staining with Alexa secondary antibodies (Invitrogen) were used. The cells were visualized using a confocal laser microscope (LSM700, Zeiss). For 3, 3′-Diaminobenzidine-nickel (DAB-Ni) staining, biotinylated secondary antibody and ABC Elite kit (Vector Labs) was performed followed by 0.02% DAB, 0.6% nickel ammonium sulfate, and 0.0045% $H_2O_2$ incubation. For DAB-Ni and HE staining, we observed the samples and took pictures with BZ-X700 (Keyence, Osaka, Japan). For DAB-Ni staining, dead cells and debris are shown in brown, and positive signals are stained in dark blue and black. The number of immunoreactive cells was quantified in every 18th section throughout the grafts and surrounding tissue and corrected by using the Abercrombie method.

**Flow cytometric analysis**. Cell aggregates were gently dissociated with Accumax into a single-cell suspension and resuspended in phenol-free, $Ca^{2+}Mg^{2+}$-free Hank's balanced salt solution (HBSS; Invitrogen) containing 2% FBS and 20 mM D-glucose (Wako). Samples were filtered through cell-strainer caps (35 μm mesh; BD Biosciences) and then subjected to surface marker staining using antibodies for PSA-NCAM, HLA-ABC, and HLA-DR. The antibodies were added and incubated at 4 °C for 20 min, and then cells were washed twice with HBSS buffer. Dead cells and debris were excluded by 7-AAD staining. For staining with Foxa2-PE antibodies, dissociated cells were fixed with 4% PFA for 30 min, washed with PBS(−) twice and re-suspended in BD Perm/Wash buffer (BD Biosciences) according to the manufacturer's protocol. The analysis was performed using a BD LSRFortessa flow cytometer, the FACSDiva software program (BD Biosciences), and FlowJo (ver.8.8.6) flow cytometry analysis software (Tree Star). HLA-ABC and HLA-DR antibodies were used to detect monkey's MHC class I and MHC class II antigens, respectively.

**Quantification by optical density of the immune-stained sections**. To quantify the results of immunostaining for MHC class II and IgG deposits, optical density was measured. IgG was deposited evenly in the immune-responded area and did not take cellular form. There were some monkeys with strong MHC class II response, which hampers accurate counting of the number of positive cells. So we decided to quantify IgG deposits and MHC class II staining with the optical density method instead of stereology. The images of the slices containing grafts were taken with BZ-X700 and analyzed by Image J (ver. 1.37; NIH, USA) software. The putamen in the section was manually defined, and its optical density was adjusted with those of the intact cortex and blank region of the slides (Fig. 7l).

**Mixed lymphocyte reaction**. We isolated PBMCs from monkey peripheral blood using Ficoll-Paque Plus (GE Lifescience) or BD Vacutainer CPT (BD) following the manufacturer's protocols. PBMCs from the animals were suspended in RPMI-1640 (GIBCO) with 10% FCS and used as responder cells. For the stimulator cells, PBMCs from the HT1 homozygous donor were treated with mitomycin C or irradiation. Then, $1 \times 10^5$ responder PBMCs and $1 \times 10^5$ stimulator cells were plated together in each well of 96-well U-bottom plates and incubated for 5 days. The plates were pulsed with 1 mCi/well of $^3$H-thymidine (GE Healthcare) for the final 8 h, and the cellular uptake of $^3$H-thymidine was quantified by a liquid scintillation counter, TRI-CARB 3100TR (Perkin Elmer). The stimulation index was calculated with the formula, S.I. = the experimental value/non-stimulated control value.

**MRI and PET**. MRI and PET studies were performed in accordance with and approved by the Animal Care and Use Committee of RIKEN Kobe Institute (MAH21-22). PET scans with $^{11}$C-PK11195 or $(S)$-$^{11}$C-KTP-Me were performed by an animal PET scanner (microPET Focus220; Siemens Medical Solutions) to identify the activation of microglia after cell transplantation. High-resolution T1-weighted and T2-weighted images were obtained using a 3T MRI scanner (MAGNETOM Verio, Siemens AG). Before MRI and PET, animals were initially sedated by an intramuscular injection of ketamine (10 mg/kg) with atropine (0.1 mg/kg) and intubated. Animals were then deeply anesthetized and maintained either by inhaled isoflurane (1.0%) during MRI or by continuous intravenous infusion of propofol (10 mg/kg/h) during PET. At the beginning of a PET scan, 37 MBq/kg of $^{11}$C-PK11195 or $(S)$-$^{11}$C-KTP-Me was injected intravenously. The scan with 3D list mode acquisition was performed for a duration of 90 min, and the obtained data were sorted and reconstructed into 4D time-frame data. The T1-weighted and T2-weighted MRI images were preprocessed for inhomogeneity correction, motion, and brain segmentation using FREESURFER[38] and NHP version of HCP pipeline[39]. $^{11}$C-PK11195 and $(S)$-$^{11}$C-KTP-Me were synthesized using $^{11}$C-$CH_3I$ according to previous reports[40, 41]. PET images obtained at each time point after transplantation (Pre, 1W, 1M, 2M, 3M) were co-registered with

MRI images obtained at the closest time point using boundary-based registration[42] and corrected for partial volume effect using a region-based voxel-wise correction[43, 44] and segmentation data. The PET data of $^{11}$C-PK11195 was estimated for the input function using a supervised algorithm (SUPERPK)[45] and analyzed for quantification of BP using a simplified reference tissue model[46, 47]. In the case of $(S)$-$^{11}$C-KTP-Me, a linearized reference tissue parametric imaging method, MRTM[48], was used to quantify BP, and the cerebellum cortex was used as the reference tissue. All images were structurally standardized into the macaque MNI space[49], and two slices were sectioned coronally and axially at $y = 0$ mm and $z = 3$ mm in ACPC space for T1w, T2w, $^{11}$C-PK11195 BP and $(S)$-$^{11}$C-KTP-Me BP. BP values are shown with a color range from 0 to 1 for $^{11}$C-PK11195 BP and from the 5th to 95th percentiles for $(S)$-$^{11}$C-KTP-Me. A total of 222 scan data (T1-weighted MRI = 56; T2-weighted MRI = 56; $^{11}$C-PK11195 PET = 56; and $(S)$-$^{11}$C-KTP-Me PET = 54) was fully automatically analyzed.

**RT-PCR**. The total RNA fraction was extracted using an RNeasy Mini kit (Qiagen) and reverse-transcribed using Super Script III First-Strand Synthesis System (Invitrogen). For PCR amplification, reactions were performed using HotStarTaq (Qiagen). Quantitative PCR reactions were carried out with Power Syber (Applied Biosystems) according to the manufacturer's instructions. The expression level of each gene was normalized to that of *GAPDH* using the ΔCT method. Genomic DNA was extracted using the DNeasy Blood and Tissue kit (Qiagen). Primers used for the reactions are shown in Supplementary Table 3.

**Statistics**. Data are expressed as means ± SD or means ± SEM, and the differences were tested by commercially available Prism 6 software (GraphPad) and SPSS software ver. 18.0 (IBM). *P*-values <0.05 were considered significant.

**Data availability**. The data that support the findings of this study are available from the corresponding author upon reasonable request.

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

## Acknowledgements

We thank Dr. Peter Karagiannis for critical reading of the manuscript, Mr. Kei Kubota for their technical assistance, and Astellas Pharma Inc. for providing Tacrolimus for this study. This study was supported by a grant from the Network Program for Realization of Regenerative Medicine from the Japan Agency for Medical Research and Development (AMED).

## Author contributions

A.Morizane and J.T. designed the study. A.Morizane, T.K., E.Y. and D.D. performed animal surgery and histological studies. H.M., S.T., T.H., A.Mawatari, H.D., M.F.G. and H.O. performed the PET and MRI study, preprocessing of the imaging data, and statistical analysis. T.S. performed genotyping of the monkeys. K.Okita and S.Y. established primate iPSCs from an MHC-homo monkey. H.I., Y.I. and K.Ogasawara designed and performed MLR experiments. A.Morizane, T.H., H.O., T.S., H.I., K.Ogasawara. and J.T. wrote the manuscript.
