## [Peer Review file · Nature Communications]

Editorial Note: This manuscript has been previously reviewed at another journal that is not operating a transparent peer review scheme. This document only contains reviewer comments and rebuttal letters for versions considered at Nature Communications. Mentions of prior referee reports have been redacted.

REVIEWERS' COMMENTS:

Reviewer #1 (Remarks to the Author):

They show that MHC-match-donor cells can reduce inflammation, leading to an increased survival of TH-positive neurons. These data are presented in a quantitative manner and now I believe the manuscript represents a significant advancement in stem cell therapy, except one point as for SFig6e.

The pictures they added do not support the author's conclusion. The author claimed that "no GFP+ grafted cells expressed Iba-1 or CD45 in vivo (Supplementary Fig. 6c, e)". However, the pictures they provided showed that some GFP+ cells look like Iba-1 positive. therefore it is difficult to believe author's claim based on their picture. How did the authors define Iba-1 signals? Are the red signals in the picture are mostly background? Could the authors provide the quantification method of IbaI with quantitative data?

7. Iba-1+ / GFP+/- picture was not shown although the authors mentioned this in the text. Please add this photo.

→ We added pictures of immunohistochemistry (GFP/Iba-I/DAPI) as Supplementary Fig. 6e and changed the legend accordingly.

REVIEWERS' COMMENTS:

Reviewer #1 (Remarks to the Author):

They show that MHC-match-donor cells can reduce inflammation, leading to an increased survival of TH-positive neurons. These data are presented in a quantitative manner and now I believe the manuscript represents a significant advancement in stem cell therapy, except one point as for SFig6e.

The pictures they added do not support the author's conclusion. The author claimed that "no GFP+ grafted cells expressed Iba-1 or CD45 in vivo (Supplementary Fig. 6c, e)". However, the pictures they provided showed that some GFP+ cells look like Iba-1 positive. therefore it is difficult to believe author's claim based on their picture. How did the authors define Iba-1 signals? Are the red signals in the picture are mostly background? Could the authors provide the quantification method of IbaI with quantitative data?

→We apologize for providing poor quality pictures of GFP/Iba-1 immunostaining. The poor quality was due to low specificity of the GFP antibody that we used previously. For the revised manuscript, we changed the GFP antibody (GFP: rat IgG, Nakalai 04404-84, clone GF090R, 1:1000) and stained the slices again. We noticed that the original GFP antibody not only stained GFP poorly, but also affected the Iba-1 staining negatively. The combination of the new GFP antibody and the same Iba-1 antibody improved the Iba-1 signal.

To quantify Iba-1 positive cells, we counted the slices of single-immunostaining with DAB-Ni enhancement (Fig.6 a, b, d).

Previous version of Supplementary Fig.6e (GFP; green, Iba-1; red, DAPI; blue)

New version of Fig. 6i (GFP; green, Iba-1; magenta, DAPI; blue)

We changed the color of the Iba-1 staining from red to magenta according to the editor's request.

As a reference, a picture of low magnified view is shown below.

GFP Iba-1 (low magnified view of Fig.6 i)

200 μ m

As we showed in Fig. 2 c, d and Supplementary Fig. 1 of the newest submission, the efficiency of GFP labeling for donor cells was not high (20% on differentiation day14).

Recent lineage tracing studies have shown that microglia originate from yolk sac erythromyeloid progenitors (EMP) generated during primitive hematopoiesis (Ginhoux et al., 2010; Kierdorf et al., 2013; Schulz et al., 2012). This year, several groups have reported protocols that induce microglia from human iPSCs. In those protocols, cells of hematopoietic lineage (CD34+, CD43+, etc) were firstly induced from iPSCs (Pandya et al., 2017; Abud et al., 2017; Douvaras et al., 2017).

In our present study, more than 99% of donor cells were positive for PSA-NCAM, indicating the cells were of neuroectodermal lineage.

Taking the above into consideration, we conclude that Iba+1+ microglia in the grafted area were derived from the host brain and not from the grafted cells.

References

1. Ginhoux, F., Greter, M., Leboeuf, M., *et al.* Fate mapping analysis reveals that adult microglia derive from primitive macrophages. *Science* **330**, 841-845 (2010)
2. Kierdorf, K., Erny, D., Goldmann, T., *et al.* Microglia emerge from erythromyeloid precursors via Pu.1- and Irf8-dependent path-ways. *Nat. Neurosci.* **16**, 273-280 (2013)
3. Schulz, C., Gomez Perdiguero, E., Chorro, L., *et al.* A lineage of myeloid cells independent of Myb and hematopoietic stem cells. *Science* **336**, 86-90 (2012)
4. Pandya, H., Shen, M. J., Ichikawa, D. M., Sedlock, A. B., *et al.* Differentiation of human and murine induced pluripotent stem cells to microglia-like cells. *Nat Neurosci* **20**, 753-759 (2017).
5. Abud, E. M., Ramirez, R. N., Martinez, E. S., Healy, L. M., *et al.* iPSC-Derived Human Microglia-like Cells to Study Neurological Diseases. *Neuron* **94**, 278-293.e9 (2017).
6. Douvaras, P., Sun, B., Wang, M., Kruglikov, I., *et al.* Directed Differentiation of Human Pluripotent Stem Cells to Microglia. *Stem Cell Reports* **8**, 1516-1524 (2017).